# County-level variations in linkage to care among people newly diagnosed with HIV in South Carolina: A longitudinal analysis from 2010 to 2018

Fanghui Shi[1,2,3]*, Jiajia Zhang[1,3,4], Chengbo Zeng[1,2,3], Xiaowen Sun[1,3,4], Zhenlong Li[3,5], Xueying Yang[1,2,3], Sharon Weissman[3,6], Bankole Olatosi[1,3,7], Xiaoming Li[1,2,3]

1 South Carolina SmartState Center for Healthcare Quality, Columbia, South Carolina, United States of America, 2 Department of Health Promotion, Education and Behavior, Arnold School of Public Health, University of South Carolina, Columbia, South Carolina, United States of America, 3 University of South Carolina Big Data Health Science Center, Columbia, South Carolina, United States of America, 4 Department of Epidemiology and Biostatistics, Arnold School of Public Health, University of South Carolina, Columbia, South Carolina, United States of America, 5 Geoinformation and Big data Research Lab, Department of Geography, College of Arts and Sciences, University of South Carolina, Columbia, South Carolina, United States of America, 6 School of Medicine, University of South Carolina, Columbia, South Carolina, United States of America, 7 Department of Health Services, Policy and Management, Arnold School of Public Health, University of South Carolina, Columbia, South Carolina, United States of America

* FSHI@email.sc.edu

**Data Availability Statement:** Data is not publicly available due to provisions in our data use agreements with state agencies/data providers,

## Abstract

### Background

Timely linkage to care (LTC) is key in the HIV care continuum, as it enables people newly diagnosed with HIV (PNWH) to benefit from HIV treatment at the earliest stage. Previous studies have found LTC disparities by individual factors, but data are limited beyond the individual level, especially at the county level. This study examined the temporal and geographic variations of county-level LTC status across 46 counties in South Carolina (SC) from 2010 to 2018 and the association of county-level characteristics with LTC status.

### Methods

All adults newly diagnosed with HIV from 2010 to 2018 in SC were included in this study. County-level LTC status was defined as 1 = "high LTC ($\geq$ yearly national LTC percentage)" and 0 = "low LTC (< yearly national LTC percentage)". A generalized estimating equation model with stepwise selection was employed to examine the relationship between 29 county-level characteristics and LTC status.

### Results

The number of counties with high LTC in SC decreased from 34 to 21 from 2010 to 2018. In the generalized estimating equation model, six out of 29 factors were significantly associated with LTC status. Counties with a higher percentage of males (OR = 0.07, 95%CI: 0.02~0.29) and persons with at least four years of college (OR = 0.07, 95%CI: 0.02~0.34)

institutional policy, and ethical requirements. We make access to such data available via approved data access requests from the IRB of the University of South Carolina (contact Lisa M. Johnson at lisaj@mailbox.sc.edu).

**Funding:** The research reported in this publication was supported by the National Institute of Allergy and Infectious Diseases of the National Institutes of Health under Award Number R01AI127203 (PI: XL) and R01AI164947 (PI: JZ,BO). This work was also partially supported by a SPARC Graduate Research Grant from the office of the Vice President for Research at the University of South Carolina (grant #: 115400-22-59203) (PI: FS). Dr. Xueying Yang's effort is supported by ASPIRE -I, TRACK-2 from the office of the Vice President for Research at the University of South Carolina (grant #: 115400-22-60028). The content is solely the responsibility of the authors and does not necessarily represent the official views of the National Institutes of Health. The funders had no role in study design, data collection and analysis, decision to publish, or preparation of the manuscript.

**Competing interests:** The authors have declared that no competing interests exist.

were less likely to have high LTC. However, counties with more mental health centers per PNWH (OR = 45.09, 95%CI: 6.81~298.55) were more likely to have high LTC.

## Conclusions

Factors associated with demographic characteristics and healthcare resources contributed to the variations of LTC status at the county level. Interventions targeting increasing the accessibility to mental health facilities could help improve LTC.

## Introduction

Timely linkage to care (LTC) is a crucial early step for treatment success in HIV control, but it remains a significant challenge for people newly diagnosed with HIV (PNWH) in South Carolina (SC) [1–3]. According to the latest established federal benchmark, timely LTC refers to the completion of a visit with an HIV healthcare provider (at least one documentation of CD4 or viral load test) within the first month (30 days) after HIV diagnosis [4]. According to the state surveillance data, there were around 748 PNWH annually in SC from 2009 to 2020 [5]. Among them, men, African Americans, people aged 20–29, and men who have sex with men were disproportionately affected by HIV, making up low percentages of SC's total population but comprising high percentages of PNWH [5]. For example, men comprise 48% of SC's total population but makeup 80% of 1,556 PNWH in SC during the two-year period 2018–2019 [5]. In 2019, only 76% of 797 PNWH were linked to care within one month in SC, which was lower than the national goal (85%) launched by the White House in 2020 [1]. In addition, it was much lower than the goal launched by the SC Department of Health and Environmental Control (SC DHEC), which aims to achieve that 90% of newly diagnosed individuals should be linked to care by December 31, 2024 [5]. More investigations on factors associated with delayed LTC are needed to provide empirical evidence for future potential interventions.

Previous studies have explored factors associated with HIV outcomes, but most have focused on the individual level. For instance, consistent findings show disparities in LTC based on race/ethnicity, gender, and age [6–10]. However, these studies do not account for important social and structural factors that may impact LTC. Understanding these factors is essential, as this could provide evidence for future efforts in policymaking and structural-level strategies to improve LTC [11–13]. Also, findings in the current literature on LTC are mixed, especially regarding structural/social factors. For instance, there are inconsistent findings regarding the impact of distance to care and transportation accessibility on LTC. Some studies show these factors do not impact LTC, but others find these as significant LTC barriers [14–17].

Previous studies have only considered a limited number of structural/social factors, which may underestimate the association of structural factors with LTC. The structural factors associated with LTC can be summarized into four dimensions based on the sociological framework: (1) demographic characteristics (e.g., racial heterogeneity, percent of poverty, and educational attainment), (2) physical characteristics (e.g., the number of mental health centers or Ryan White HIV centers, the primary care provider rate), (3) social characteristics (e.g., violent crime, religious adherence, and social capital), and (4) health behaviors (e.g., smoking and excessive drinking) [15–20]. To our best knowledge, a dearth of studies on LTC incorporates all these four dimensions of structural predictors. The current study aimed to investigate the relationship between county-level factors and LTC among PNWH in SC when considering multi-dimensional structural factors.

## Methods

### Data sources and linkage

The study population included all people (aged ≥18) newly diagnosed with HIV from January 2010 to December 2018 across 46 counties in SC. Individual de-identified laboratory reports of CD4 counts and viral load were extracted from the enhanced HIV/AIDS reporting system (eHARS) in the SC DHEC [21]. They were used to calculate the county-level timely LTC percentage based on the Centers for Disease Control and Prevention (CDC) definition [4].

First, county-level variables were extracted from multiple public database sources with the Federal Information Processing Standards (FIPS) as the identification of each county, including the American Community Survey (ACS), County Health Rankings & Roadmaps, and the US Congress Joint Economic Committee. According to the census data user guide, the ACS 5-year estimates data were used since multi-year estimates could increase statistical reliability for small population groups [22]. Then, county-level LTC data and all county-level factors were linked by FIPS code and calendar year. The Institutional Review Boards at the University of South Carolina and SC DHEC approved the study protocol (#Pro00068124). The IRB approved this study as a non-human subject study, and no participant consent is needed.

### County-level LTC status

According to CDC, timely LTC was measured by records of ≥ 1 CD4 (count or percentage) or viral load tests performed within one month after HIV diagnosis, including tests performed on the same date as the date of diagnosis [4]. Based on this definition, we classified the individual-level LTC status as "timely LTC" and "delayed LTC." The county-level timely LTC percentage was calculated as the number of "timely LTC" divided by the number of newly diagnosed HIV cases for each county in the specified calendar year. By comparing the yearly county-level LTC percentage to the yearly national LTC percentage in the US from 2010 to 2018 (70.2%, 70.4%, 71.4%, 72.6%, 74.5%, 75%, 75.9%, 78.3%, and 80.2%) [23], we defined county-level LTC status as 1 = "high LTC (≥ national LTC percentage)" or 0 = "low LTC (< national LTC percentage)" (reference group). According to the technical notes from CDC NCHHSTP AtlasPlus, national LTC is presented for persons aged ≥ 13 years and only for states with complete laboratory data (at least 95% of laboratory results are reported to the surveillance programs and transmitted to the CDC). From 2010 to 2018, the calculation of national LTC percentage ranges from 14 to 43 jurisdictions [24]. The list of jurisdictions for which data are presented by year is presented in S1 Table.

### County-level variables

We included county-level variables that are publicly available from multiple datasets or aggregated from individual-level EHR data, and these factors were organized into four dimensions: demographic, physical, social characteristics, and health behaviors [15]. A total of 29 county-level factors were included in this study, and detailed information (e.g., definition, data source, and years of data used) for each variable is provided in Table 1. All missing data from 2010 to 2018 were imputed using the information from the neighboring year.

### Sociodemographic characteristics

County-level sociodemographic information refers to the population's demographic composition and broad socioeconomic characteristics in a local area [25]. For demographic composition, eight variables were considered. Four variables were extracted from ACS 5-year estimates, including population size, male (%), age (≥18 years, %), and Black (%). For each

**Table 1. The detailed definition, data source, and years of data extracted for each county-level variable.**

| Variables | Definitions | Year |
|---|---|---|
| **Sociodemographic characteristics** | | |
| Population size[a] | Total weighted population | 2010–2018 |
| Blacks among persons newly diagnosed with HIV (%)[b] | Percent of Black persons among people newly diagnosed with HIV each year | 2010–2018 |
| Males among persons newly diagnosed with HIV (%)[b] | Percent of male persons among people newly diagnosed with HIV each year | 2010–2018 |
| Male (%)[a] | Percent of male persons | 2010–2018 |
| Age ($\geq$18, %)[a] | Percent of persons aged > = 18 years old | 2010–2018 |
| Black (%)[a] | Percent of Black persons | 2010–2018 |
| High education (%)[a] | Percent of 25 years and older persons with at least four years of college | 2010–2018 |
| Low education (%)[a] | Percent of 25 years and older persons with less than a high school education | 2010–2018 |
| Vacant houses (%)[a] | Percent of vacant houses in high SES neighborhoods in addition to abandoned housing | 2010–2018 |
| Poverty (%)[a] | Percent of 18–64 years old persons living below the federally defined poverty line | 2010–2018 |
| Median income ($)[a] | Annual median household income | 2010–2018 |
| No insurance (%)[a] | Percent of persons with no health insurance coverage | 2010–2018 |
| Public assistance (%)[a] | Percent of households with public assistance | 2010–2018 |
| Unemployed (%)[a] | Percent of 16 years and older persons who are unemployed | 2010–2018 |
| No transportation (%)[a] | Percent of occupied housing units without access to a vehicle | 2010–2018 |
| White/non-White residential segregation index[c] | The percentage of either White or non-White residents that would have to move to different geographic areas to produce a distribution that matches that of the larger area | 2016–2018 |
| Black/White residential segregation index[c] | The percentage of either Black or White residents that would have to move to different geographic areas to produce a distribution that matches that of the larger area | 2016–2018 |
| **Physical characteristics** | | |
| primary care providers[a] | Number of primary care providers per 100,000 population | 2010–2018 |
| Ryan White HIV centers[d] | Number of Ryan White HIV centers per newly diagnosed HIV case each year within 25 miles radius | 2010–2018 |
| Mental health centers[d] | Number of mental health centers per newly diagnosed HIV case each year within 25 miles radius | 2010–2018 |
| **Social characteristics** | | |
| Gini index [a] | Income inequality represented by statistical measure of income dispersion | 2010–2018 |
| Religious adherence (%) [a] | Percent of persons with religious adherence | 2010 |
| Family unity [e] | The share of births that are to unwed mothers, children living in single-parent families, and women aged 35–44 who are married | 2018 |
| Community health[e] | Non-religious non-profits per capita, congregations per capita, and the informal civil society subindex | 2018 |

(*Continued*)

**Table 1.** (Continued)

| Variables | Definitions | Year |
|---|---|---|
| Institution health[e] | Presidential voting rate, census response rate, and confidence subindex | 2018 |
| Collective efficiency[e] | Violent crimes per 100,000 people | 2018 |
| **health behaviors** | | |
| Smoking (%)[c] | Percent of adults who are current smokers | 2011–2018 |
| Drinking (%)[c] | Percent of adults reporting binge or heavy drinking | 2011–2018 |

[a] Extracted from American Community Survey 5-year Estimate

[b] Aggregated from individual-level enhanced HIV/AIDS reporting system (eHARS) in SC DHEC

[c] Extracted from County Health Rankings & Roadmaps

[d] Extracted from US Department of Health and Human Services (DHHS) Data Warehouse and health department websites in SC and its neighboring states

[e] Extracted from US Congress Joint Economic Committee

calendar year from 2010 to 2018, the 5-year estimates refer to data collected over the past five years. For example, in 2018, the 5-year estimation refers to data collected from 2014 to 2018. Two demographic compositions of PNWH, including the percent of Black persons among PNWH and the percent of Male persons among PNWH each year, were aggregated and calculated based on the individual level race and gender data from eHARS. Two segregation indices, including the White/non-White residential segregation index and the Black/White residential segregation index, were extracted from County Health Rankings & Roadmaps. The residential segregation index, ranging from 0 to 100, can be interpreted as the percentage of one racial group that have to move to a different geographic area (census tract) to produce a distribution that of the larger area (county). The higher the residential segregation index score, the greater the residential segregation between two racial groups [26].

For socioeconomic characteristics, nine variables were extracted from ACS (5-year estimates), including the percentage of persons aged over 25 years old with less than high school education (lower education), the percentage of persons aged over 25 years old with at least four years of college (higher education), the proportion of people aged 18–64 years living in below the federally defined poverty line, the proportion of household with public assistance income, median household annual income, percentage of no health insurance coverage, unemployment rate, percentage of vacant homes in neighborhoods with high socioeconomic status (SES) in addition to abandoned housing, and transportation accessibility (proportion of occupied housing units without access to a vehicle) [27–29].

## Physical characteristics

Physical characteristics represent the accessibility of social settings in the built environment and relevant social resources. Three factors, including the number of primary care providers per 100,000 people based on US Health data, the number of Ryan White HIV centers per PNWH, and the number of mental health centers per PNWH within 25 miles radius of each county in SC, were used to reflect local people's access to health care access opportunity [30, 31].

**Social characteristics.** Social characteristics refer to social networks and social culture-related characteristics that are related to inequities or social disorganization [32, 33]. We

included one factor about income inequalities (GINI index), one factor about the religious environment, and four factors about social capital. GINI index—a measure of income inequality between 0 and 1, with 0 being complete equality and 1 being complete inequality—was extracted from ACS, and the religious environment was measured by the proportion of religious adherents based on US Religious Data. Social capital factors included four variables, namely community health, institutional health, family unity, and collective efficacy, extracted from the 2018 US Congress Joint Economic Committee [34]. The detailed procedure for creating the former three indices (community health, institutional health, and family unity) is described elsewhere, and these factors were coded that higher scores corresponded with higher social capital levels [34, 35]. Collective efficacy was measured by the number of reported violent crimes per 100,000 population.

**Health behaviors.** Health behaviors refer to actions individuals take that may affect their health. County-level health behaviors, including excessive drinking and adult smoking, were extracted from CHRR. Excessive drinking was measured by the percentage of adults reporting binge or heavy drinking in the past 30 days. Adult smoking was calculated by the percentage of adults who are current smokers.

## Statistical analysis

First, spatial-temporal distribution and variation of yearly LTC status were described by nine geographic maps of LTC percentage differences between the county and national levels from 2010 to 2018. The 46 counties in the nine maps were further grouped based on four Public Health Regions in SC, including Upstate, Midlands, Pee Dee, and Lowcountry [36]. Second, LTC percentages across 46 counties from 2010 to 2018 were illustrated using a heat map. Third, descriptive statistics were reported for all the county-level variables, including the 25th percentile, median, 75th percentile, and Interquartile Range (IQR). Fourth, we used longitudinal data from 2010 to 2018 to fit a Generalized Estimating Equation (GEE) model with stepwise selection to explore the relationship between county-level characteristics and LTC status. The stepwise selection is a procedure where we fit our regression model from a set of candidate variables by entering and removing variables based on the cut point of the p-value being 0.2 [37]. The exchangeable correlation structure within counties was used for the GEE approach to account for the repeated measure of county-level information. All analyses were conducted using R version 4.0.3, except for the geographic map created using GeoPandas. The significant level of statistical results was set at a P-value of 0.05.

## Results

### Descriptive statistics

The yearly number of adult PNWH was 746, 739, 687, 709, 760, 699, 775, 760, and 749 in SC from 2010 to 2018, respectively. (S1 Fig) The number of counties with a high LTC decreased from 34 (73.91%) in 2010 to 21 (45.65%) in 2018. (Table 2) However, the state average timely LTC percentage in SC is relatively stable, with the timely LTC percentage being 78.55% in 2010 and 80.99% in 2018. Additionally, the percentage of linkage to care within 60 days was 88.07% in 2010 and 88.49% in 2018. 90.75% and 90.63% of all PNWH were linked to care within 90 days after diagnosis in 2010 and 2018, respectively. (S2 Fig).

Table 3 describes the distribution of county-level characteristics across 46 counties in SC, and only data in the years 2010, 2014, and 2018 were described due to limited table space. In over 25% of the counties, 100% of PNWH were Black in 2010, but this percentage decreased to 88.89% in 2018. Half of the counties consistently have more than 48% males, more than 32% Blacks, and more than 76% aged above 18 years. Across the counties, there were relatively

**Table 2. Linkage to care status across 46 counties in South Carolina, n (%).**

| Linkage to care status | Below the jurisdictive national level | Above the jurisdictive national level |
|---|---|---|
| 2010 | 12 (26.09%) | 34 (73.91%) |
| 2011 | 10 (21.74%) | 36 (78.26%) |
| 2012 | 14 (30.43%) | 32 (69.57%) |
| 2013 | 14 (30.43%) | 32 (69.57%) |
| 2014 | 19 (41.30%) | 27 (58.70%) |
| 2015 | 21 (45.65%) | 25 (54.35%) |
| 2016 | 16 (34.78%) | 30 (65.22%) |
| 2017 | 17 (36.96%) | 29 (63.04%) |
| 2018 | 25 (54.35%) | 21 (45.65%) |

large variations in the proportion of Black persons (25th percentile: 24%, 75th percentile: 7%) and religious adherence (25th percentile: 41.2%, 75th percentile: 59.4%), with the IQRs over 15% in 2010, 2014, and 2018. In contrast, there were relatively more minor variations across the counties for the percent of lower/higher education attainment, poverty, vacant homes, transportation accessibility, no insurance coverage, and smoking/drinking behaviors, with the IQRs ranging from 2% to 10%.

## LTC status across counties in SC

Figs 1 and 2 illustrate the spatiotemporal variations of LTC across 46 counties, and county-level disparities in LTC were identified. The timely LTC percentage in some Upstate counties, including Greenville and Anderson, was consistently higher than the national level. In contrast, some Midlands (e.g., Edgefield, Saluda, Chester, and Lexington) and Lowcountry (e.g., Allendale and Bamberg) counties had low LTC in at least six years from 2010 to 2018.

## Generalized estimating equation model with stepwise selection

Table 4 shows that after stepwise selection, 9 out of 29 county-level variables were retained in the final GEE model. Six variables were significantly associated with LTC status after including nine variables in the adjusted model. Counties with high LTC in SC decreased from 2010 to 2018 (OR: 0.87, 95%CI: 0.80~0.95). For demographic characteristics-related factors, the proportion of male persons (OR = 0.07, 95%CI: 0.02~0.29) and the proportion of high education (OR = 0.07, 95%CI:0.02~0.34) was negatively associated with high LTC. In addition, living in a county with a larger ratio of mental health centers per PNWH was related to a higher likelihood of high LTC (OR = 45.09, 95%CI:6.81~298.55). Among social characteristics-related factors, the number of violent crimes per 100,000 people was positively associated with high LTC (OR = 4.86, 95% CI: 1.43~16.59).

## Discussion

This study described both the temporal and spatial variations of LTC status across 46 counties in SC from 2010 to 2018 and investigated the relationship between county-level characteristics and these variations. Twenty-nine county-level variables across demographic, physical, social characteristics and health behaviors domains were selected, and six of them were detected to be significantly associated with LTC status.

There were apparent spatial disparities in LTC percentage in SC, with some counties constantly having lower or higher LTC percentages than the national level. Generally, compared

**Table 3. Descriptive statistics of county-level variables.**

| Predictors | 25th percentile | Median | 75th percentile | Interquartile Range (IQR) |
|---|---|---|---|---|
| **Demographic characteristics** | | | | |
| Population size | | | | |
| 2010 | 27282 | 57499 | 133577 | 106295 |
| 2014 | 27003 | 58048 | 141594 | 114590 |
| 2018 | 27259 | 59158 | 151246 | 123987 |
| Blacks among persons newly diagnosed with HIV (%) | | | | |
| 2010 | 66.67 | 82.58 | 100.00 | 0.33 |
| 2014 | 50.54 | 66.67 | 88.89 | 0.38 |
| 2018 | 50.61 | 75.00 | 85.58 | 0.35 |
| Male among persons newly diagnosed with HIV (%) | | | | |
| 2010 | 60.00 | 76.51 | 87.96 | 0.28 |
| 2014 | 75.00 | 83.33 | 100.00 | 0.25 |
| 2018 | 71.43 | 80.63 | 97.06 | 0.26 |
| Male (%) | | | | |
| 2010 | 48.25 | 48.58 | 49.58 | 1.33 |
| 2014 | 47.85 | 48.57 | 49.26 | 1.41 |
| 2018 | 47.85 | 48.54 | 49.26 | 1.41 |
| Black (%) | | | | |
| 2010 | 24.99 | 33.51 | 47.42 | 22.43 |
| 2014 | 24.76 | 33.23 | 46.66 | 21.90 |
| 2018 | 23.84 | 32.2 | 47.02 | 23.18 |
| Age (≥18, %) | | | | |
| 2010 | 75.12 | 76.22 | 76.95 | 1.83 |
| 2014 | 76.08 | 77.28 | 78.81 | 2.73 |
| 2018 | 76.69 | 78.04 | 79.78 | 3.09 |
| Low education (%) | | | | |
| 2010 | 17.39 | 21.74 | 24.68 | 7.29 |
| 2014 | 15.32 | 19.33 | 21.84 | 6.52 |
| 2018 | 13.35 | 16.92 | 19.21 | 5.86 |
| High education (%) | | | | |
| 2010 | 12.88 | 16.52 | 21.81 | 8.93 |
| 2014 | 13.03 | 18.06 | 22.41 | 9.38 |
| 2018 | 14.43 | 18.59 | 24.49 | 10.06 |
| Poverty (%) | | | | |
| 2010 | 14.21 | 17.31 | 19.55 | 5.34 |
| 2014 | 16.60 | 20.38 | 22.90 | 6.30 |
| 2018 | 15.43 | 18.35 | 20.84 | 5.41 |
| Median income ($) | | | | |
| 2010 | 33066 | 38588 | 42871 | 9805 |
| 2014 | 33615 | 38610 | 43203 | 9588 |
| 2018 | 36276 | 42514 | 49392 | 13116 |
| Public assistance (%) | | | | |
| 2010 | 1.36 | 1.58 | 2.04 | 0.68 |
| 2014 | 1.27 | 1.65 | 2.06 | 0.79 |
| 2018 | 1.22 | 1.38 | 1.76 | 0.54 |
| Vacant house (%) | | | | |
| 2010 | 12.79 | 16.67 | 20.69 | 7.90 |

(*Continued*)

**Table 3.** (*Continued*)

| Predictors | 25[th] percentile | Median | 75[th] percentile | Interquartile Range (IQR) |
|---|---|---|---|---|
| 2014 | 13.30 | 16.45 | 20.78 | 7.48 |
| 2018 | 13.30 | 17.74 | 22.78 | 9.48 |
| Transportation accessibility (%) | | | | |
| 2010 | 6.52 | 8.52 | 10.32 | 3.80 |
| 2014 | 6.30 | 8.04 | 10.01 | 3.71 |
| 2018 | 6.04 | 7.29 | 9.87 | 3.83 |
| No insurance coverage (%) | | | | |
| 2010 | 15.50 | 16.95 | 18.70 | 3.20 |
| 2014 | 15.17 | 16.34 | 18.26 | 3.09 |
| 2018 | 10.09 | 11.33 | 12.18 | 2.09 |
| White/non-White residential segregation index[a] | 24.67 | 30.00 | 35.33 | 10.66 |
| Black/White residential Segregation index[a] | 26.67 | 30.83 | 38.67 | 12.00 |
| **Physical characteristics** | | | | |
| Primary care providers | | | | |
| 2010 | 61.70 | 78.76 | 105.93 | 44.23 |
| 2014 | 35.78 | 48.12 | 58.93 | 23.15 |
| 2018 | 37.44 | 47.56 | 68.25 | 30.81 |
| Ryan White HIV centers | | | | |
| 2010 | 0.03 | 0.15 | 0.33 | 0.31 |
| 2014 | 0.03 | 0.11 | 0.39 | 0.36 |
| 2018 | 0.03 | 0.15 | 0.47 | 0.44 |
| Mental health centers | | | | |
| 2010 | 0.15 | 0.41 | 1.00 | 0.85 |
| 2014 | 0.16 | 0.33 | 0.79 | 0.63 |
| 2018 | 0.16 | 0.38 | 1.00 | 0.84 |
| **Social characteristics** | | | | |
| Gini index | | | | |
| 2010 | 0.44 | 0.45 | 0.47 | 0.03 |
| 2014 | 0.45 | 0.46 | 0.48 | 0.03 |
| 2018 | 0.45 | 0.47 | 0.49 | 0.03 |
| Religious adherence (%)[b] | 41.2 | 53.6 | 59.4 | 18.2 |
| Family unity [b] | -1.81 | -1.16 | -0.51 | 1.30 |
| Community health [b] | -0.79 | -0.55 | -0.31 | 0.48 |
| Institution health [b] | -0.05 | 0.25 | 0.38 | 0.43 |
| Collective efficiency [b] | 406.70 | 503.00 | 629.20 | 222.50 |
| **Health behaviors** | | | | |
| Smoking (%)[c] | | | | |
| 2010 | 0.21 | 0.23 | 0.26 | 0.05 |
| 2014 | 0.19 | 0.21 | 0.23 | 0.01 |
| 2018 | 0.17 | 0.19 | 0.20 | 0.03 |
| Drinking (%)[c] | | | | |
| 2010 | 0.11 | 0.13 | 0.15 | 0.04 |
| 2014 | 0.11 | 0.12 | 0.15 | 0.04 |

(*Continued*)

**Table 3.** (Continued)

| Predictors | 25th percentile | Median | 75th percentile | Interquartile Range (IQR) |
|---|---|---|---|---|
| 2018 | 0.15 | 0.16 | 0.17 | 0.03 |

Notes:

[a] Variables were only available since 2016 and data from 2010 to 2015 were imputed using data from 2016 throughout the analysis

[b] Variables were only available in one year and were used as constant variables throughout the analysis

[c] Variables were only available since 2011, and data in 2010 were imputed using data from 2011 throughout the analysis

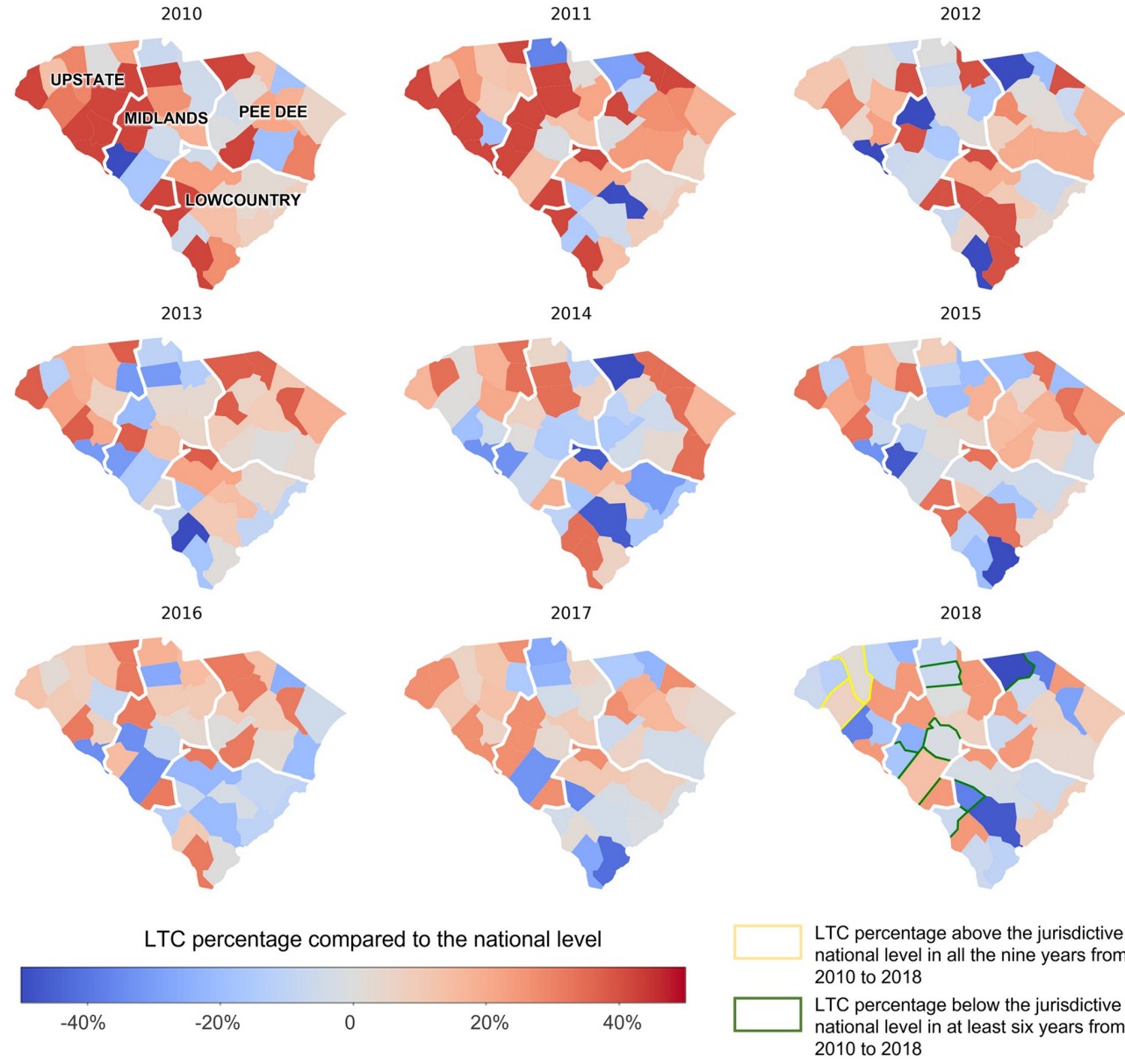

**Fig 1. Linkage to care percentage differences between county level and jurisdictive national level among people living with HIV across 46 counties in South Carolina from 2010 to 2018.**

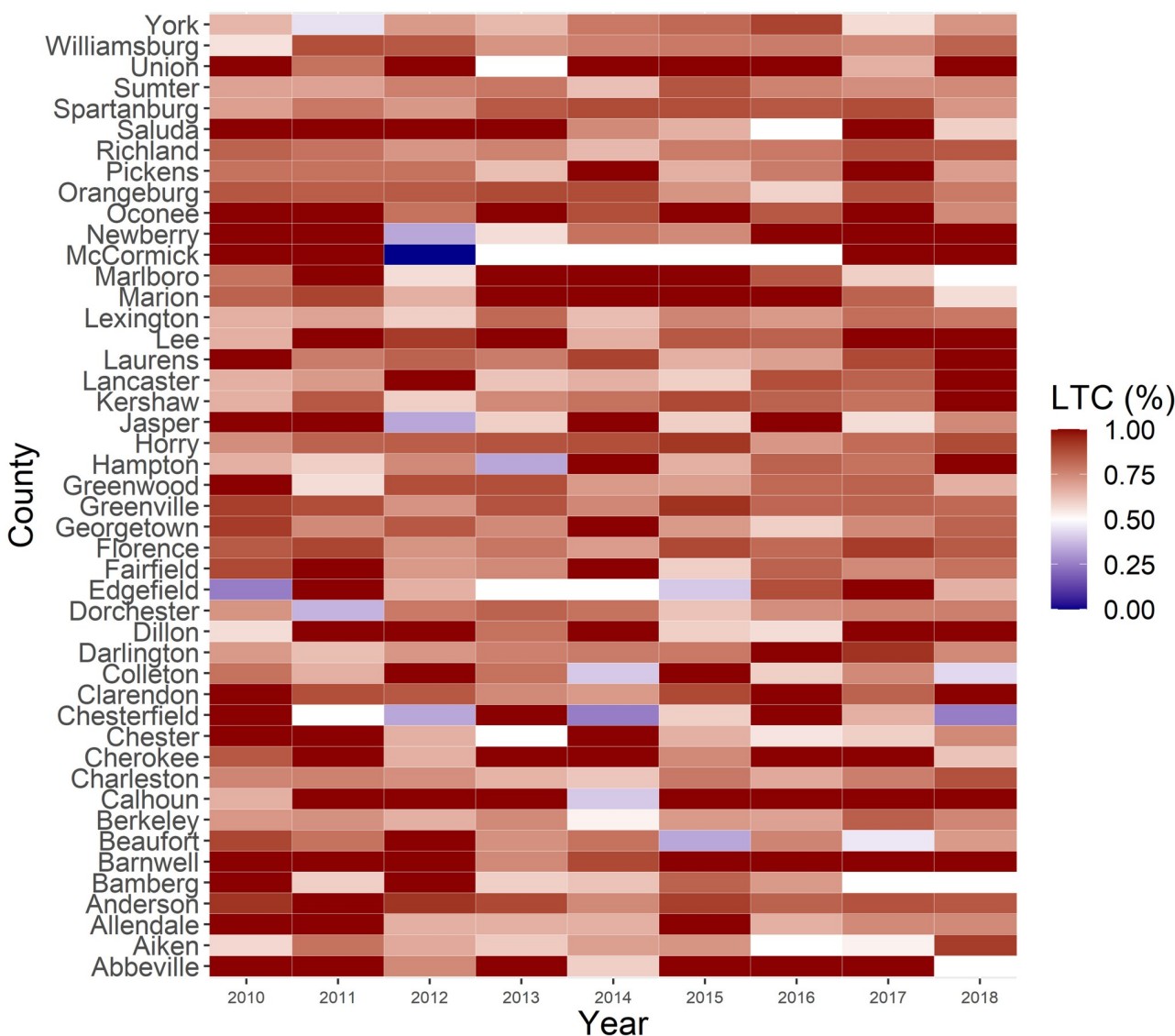

**Fig 2. Heatmap of linkage to care percentage among people living with HIV across 46 counties in South Carolina from the year 2010 to 2018.**

to the counties in the Lowcountry and Midlands region, the Upstate area tended to have high LTC. According to the epidemic profile 2020 of HIV and AIDS in Upstate, as of December 31, 2019, the Upstate has the highest number and proportion (33%) of people living with HIV in SC [36]. Despite the large prevalence, LTC efforts have improved in Upstate, with various programs and outlets for LTC [36]. In the Lowcountry region, Bamberg had the second-highest number of newly diagnosed HIV cases [38]. However, it has reported low LTC compared to the national level for at least six years from 2010 to 2018. This highlights the urgency and significance of interventions to improve LTC status in these counties [39].

Among demographic characteristics, we found a significant and negative association between the percent of the Male population and LTC status in SC, and this association persisted after controlling the percentage of PNWH who are male in the model. This finding was consistent with previous individual-level research, in which male persons were less likely

**Table 4. The association of county-level factors with linkage to care status across the counties in South Carolina from 2010 to 2018: Stepwise based Generalized Estimating Equations (GEE) model.**

| Factors | Crude OR (95%CI) | Adjusted OR (95%CI) |
|---|---|---|
| Year | 0.88 (0.81~0.95) | **0.87 (0.80~0.95)** |
| Blacks among persons newly diagnosed with HIV (%) | 0.79 (0.35~1.77) | - |
| Male among persons newly diagnosed with HIV (%) | 0.62 (0.24~1.58) | - |
| **Sociodemographic characteristics** | | |
| Population size | 2.22 (0.92~5.35) | **5.17 (1.40~19.16)** |
| Male (%) | **0.14 (0.05~0.41)** | **0.07 (0.02~0.29)** |
| Age (≥18, %) | **0.23 (0.06~0.91)** | - |
| Black (%) | 0.71 (0.32~1.67) | - |
| Low education (%) | 1.51 (0.52~4.44) | - |
| High education (%) | 0.62 (0.24~1.57) | **0.07 (0.02~0.34)** |
| Median income ($) | 0.47 (0.17~1.33) | - |
| Public assistance (%) | 0.94 (0.28~3.21) | - |
| Unemployment (%) | 3.38 (0.81,14.16) | - |
| Transportation accessibility (%) | 1.24 (0.36~4.25) | 0.19 (0.04~1.01) |
| Poverty (%) | 0.93 (0.25~3.42) | - |
| Vacant house (%) | 1.33 (0.47~3.70) | - |
| No insurance coverage (%) | **3.76 (1.23~11.50)** | - |
| White/non-White residential segregation index | **2.53 (1.08~5.96)** | 0.15 (0.00~8.85) |
| Black/White residential Segregation index | **2.76 (1.25~6.95)** | 29.07 (0.57~1478.21) |
| **Physical characteristics** | | |
| Ryan White HIV centers | 0.63 (0.19~2.00) | - |
| Mental health centers | 2.32 (0.71~7.58) | **45.09 (6.81~298.55)** |
| Primary care providers | **5.49 (1.04~28.96)** | - |
| **Social characteristics** | | |
| Gini index | 0.92 (0.22~3.84) | - |
| Religious adherence (%) | **3.36 (1.27~8.90)** | - |
| Family unity | 0.98 (0.43~2.24) | - |
| Community health | 0.27 (0.65~1.10) | - |
| Institution health | 1.04 (0.38~2.84) | - |
| Collective efficacy | **3.90 (1.32~11.50)** | **4.86 (1.43~16.59)** |
| **Health behaviors** | | |
| Smoking (%) | 1.98(0.58~6.75) | - |
| Drinking (%) | 0.37(0.13~1.01) | - |

Notes:

OR: Odds Ratio. CI: Confidence Interval.

-: Variables were not selected by the stepwise selection. All OR in bold means statistically significant

to be linked to care timely [40–42]. One possible explanation is that a high proportion of male persons may be related to masculinity norms in the local area, especially in the Deep South States [42]. Traditional masculinity ideology deters males' perception of the risks of HIV to their health and ultimately deters their health-seeking behaviors [11]. Men were disproportionately affected by HIV/AIDS, and they were likely to be influenced by the atmosphere of masculinity culture [11]. These results warrant intensified intervention efforts among male PNWH and in counties with a high proportion of male persons when promoting LTC.

The number of mental healthcare centers within a 25-mile radius of each county per PNWH was found to be positively associated with LTC status. Previous studies have found various HIV-associated mental health problems (e.g., stigma, depression, anxiety, and fear) were significant barriers to timely LTC [43–45]. In a US sample of PNWH, depression was a statistically significant predictor of failed linkage to care within three months after initial HIV diagnosis [44]. This emphasized the potential need for integration of mental health services alongside interventions at the early stage of the HIV continuum of care, such as immediately after HIV diagnosis and when initiating contact with treatment services. As one significant aspect of accessible healthcare facilities, accessible mental health centers provide PNWH with psychological counselling and services for mental health treatment. Our findings implied that interventions aimed at counties with limited mental healthcare resources might promote county-level LTC.

This study is innovative in leveraging multiple public datasets, incorporating many county-level predictors, and applying longitudinal models to investigate the county-level variations of LTC status in SC. However, there are still some limitations that need to be acknowledged. First, some counties only have a few new HIV cases, limiting the statistical power of detecting potential predictors. Second, more potential structural predictors (e.g., HIV-related discrimination and structural racism in incarceration) should be included in future studies, which are not included in the current study due to the unavailability of data. Third, individual-level factors were not included in this study. More investigations on the accumulative impacts of individual and structural factors can provide more insights into the barriers and facilitators of LTC. Fourth, there may be the modifiable areal unit problem (MAUP) since county-level data were used in the analysis. We need to be cautious when generating findings of the current study to other administrative units, such as the census tract.

## Conclusion

Considering the unsatisfactory results of LTC status in SC when compared to the national level and the concentration of low LTC percentages in counties with large HIV cases, more efforts on promoting LTC are still needed to curb the HIV epidemic. Counties with a large proportion of male persons require intensive attention, and actions that focus on improving accessible mental healthcare centers tend to be effective interventions. To get a more thorough understanding of the structural/social determinants of the LTC percentage, the effectiveness of interventions based on these factors should be evaluated. More significant country-level factors that are unavailable at present should be measured and incorporated into future studies.

## Supporting information

**S1 Fig. Linkage to care within 30 days, 60 days, or 90 days of HIV diagnosis in South Carolina, 2010 through 2018.**
(TIF)

**S2 Fig. The number of newly diagnosed HIV cases in South Carolina, 2010 through 2018.**
(TIF)

**S3 Fig. Bivariate map on Log odds of linkage to care status and the number of mental health centers per person newly diagnosed with HIV (PNWH) across South Carolina in 2018.**
(TIF)

**S1 Table. Jurisdiction(s) meeting National HIV surveillance system laboratory reporting requirements, 2010–2018.**
(DOCX)

## Acknowledgments

The authors thank the SC Department of Health and Environmental Control (DHEC), the Office of Revenue and Fiscal Affairs (RFA), and other SC agencies for contributing the data in South Carolina.

## Author Contributions

**Conceptualization:** Fanghui Shi, Jiajia Zhang, Chengbo Zeng.

**Data curation:** Fanghui Shi, Xiaowen Sun.

**Formal analysis:** Xiaowen Sun.

**Funding acquisition:** Fanghui Shi, Jiajia Zhang, Xueying Yang, Bankole Olatosi, Xiaoming Li.

**Methodology:** Jiajia Zhang.

**Software:** Zhenlong Li.

**Supervision:** Jiajia Zhang.

**Visualization:** Zhenlong Li.

**Writing – original draft:** Fanghui Shi.

**Writing – review & editing:** Fanghui Shi, Jiajia Zhang, Chengbo Zeng, Xiaowen Sun, Zhenlong Li, Xueying Yang, Sharon Weissman, Bankole Olatosi, Xiaoming Li.

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
