## [Decision Letter · Decision Letter 0]

30 Jan 2023

PONE-D-22-15791County-level determinants of linkage to care among people living with HIV in South Carolina: A longitudinal analysis from 2010 to 2018PLOS ONE

Dear Fanghui Shi,

Thank you for submitting your manuscript to PLOS ONE. After careful consideration, we feel that it has merit but does not fully meet PLOS ONE’s publication criteria as it currently stands. Therefore, we invite you to submit a revised version of the manuscript that addresses the points raised during the review process.

We look forward to receiving your revised manuscript.

Kind regards,

Csaba Varga, DVM MSc PhD

Academic Editor

PLOS ONE

and https://journals.plos.org/plosone/s/file?id=ba62/PLOSOne_formatting_sample_title_authors_affiliations.pdf.

“The research reported in this publication was supported by the National Institute of Allergy and Infectious Diseases of the National Institutes of Health under Award Number R01AI127203 and R01AI164947. This work was also partially supported by a SPARC Graduate Research Grant from the office of the Vice President for Research at the University of South Carolina (grant #: 115400-22-59203). Dr. Xueying Yang’s effort is supported by ASPIRE -I, TRACK-2 from the office of the Vice President for Research at the University of South Carolina (grant #: 115400-22-60028). The content is solely the responsibility of the authors and does not necessarily represent the official views of the National Institutes of Health. Both NIAID and NIH had no role in the design of the study, collection, analysis, and interpretation of the data.”

“The research reported in this publication was supported by the National Institute of Allergy and Infectious Diseases of the National Institutes of Health under Award Number R01AI127203 (PI: XL) and R01AI164947 (PI: JZ,BO). This work was also partially supported by a SPARC Graduate Research Grant from the office of the Vice President for Research at the University of South Carolina (grant #: 115400-22-59203) (PI: FS). Dr. Xueying Yang’s effort is supported by ASPIRE -I, TRACK-2 from the office of the Vice President for Research at the University of South Carolina (grant #: 115400-22-60028). The content is solely the responsibility of the authors and does not necessarily represent the official views of the National Institutes of Health. The funders had no role in study design, data collection and analysis, decision to publish, or preparation of the manuscript.”

7. We note that [Figure 1) #] in your submission contain [map/satellite] images which may be copyrighted. All PLOS content is published under the Creative Commons Attribution License (CC BY 4.0), which means that the manuscript, images, and Supporting Information files will be freely available online, and any third party is permitted to access, download, copy, distribute, and use these materials in any way, even commercially, with proper attribution. For these reasons, we cannot publish previously copyrighted maps or satellite images created using proprietary data, such as Google software (Google Maps, Street View, and Earth). For more information, see our copyright guidelines: http://journals.plos.org/plosone/s/licenses-and-copyright.

Natural Earth (public domain): http://www.naturalearthdata.com/.

Reviewers' comments:

Reviewer's Responses to Questions

**Comments to the Author**

1. Is the manuscript technically sound, and do the data support the conclusions?

Reviewer #1: Yes

Reviewer #2: Partly

Reviewer #3: Yes

2. Has the statistical analysis been performed appropriately and rigorously? 

Reviewer #1: Yes

Reviewer #2: No

Reviewer #3: Yes

3. Have the authors made all data underlying the findings in their manuscript fully available?

Reviewer #1: No

Reviewer #2: No

Reviewer #3: No

4. Is the manuscript presented in an intelligible fashion and written in standard English?

Reviewer #1: Yes

Reviewer #2: Yes

Reviewer #3: Yes

5. Review Comments to the Author

Reviewer #1: This is a well-written manuscript describing a county-level analysis of predictors of linkage to care among people living with HIV in South Carolina. Overall, the paper is clearly presented. I only have a few minor comments and questions.

The outcome of interest was a viral load or CD4 measurement within one month of HIV diagnosis. i recognize that this is based on federal recommendations, but at the very least a descriptive presentation of time to viral load or CD4 count would be useful. For example, how many occurred >30 days but <60 days or <90 days?

Methods, line 99: should this be: "Successful linkage to care" divided by the number of newly diagnosed HIV cases? I.e., addition of 'newly' to this sentence?

Methods, line 126: It seems as though the analysis is conducted at the county level, but this line refers to data at the census tract level. More information about how these data are aggregated up to the county level would be useful.

Results, lines 188-195: This paragraph is difficult to follow and would benefit from editing for clarity.

Table 3: Consider adding one to two decimal places to the count of Ryan White HIV centers and mental health centers per 100,000 people. Also, there is a minor typo here ("Pyan White" instead of "Ryan White").

Results, lines 201-206: Lancaster and Sumter counties are repeated in this list.

The authors have used statistical methods to identify predictors of linkage to care; however, they have not conducted a causal analysis. Thus, use of words like 'determinants' is unwarranted. In most of the Discussion, 'predictors' is used. However, in the title and in the first paragraph of the Discussion, the word 'determinants' is used. The authors are encouraged to reconsider this word usage so as not to overstate the findings.

Reviewer #2: Overall, the authors are bringing to light the social and environment factors that might be contributing to patterns in linkage to care in the state of South Carolina. This is helpful information to learn about, however, there are several items that have me hesitant to accept for publication. Please see attached document for comments.

Reviewer #3: This manuscript examines factors that influence linkage to HIV care in South Carolina at the county level between 2010 and 2018. The paper is very well written and clearly and concisely summarized. Overall, the study is well done and don't have any major concerns, but numerous minor issues should be addressed before publication consideration.

Minor comments:

-Please provide the IRB IDs approved bu SC and SC DHEC in text (line 93).

-Please improve the resolution of figures 1 and 2.

-Figure 1: Please add a white halo around the labels in the 2010 map. Overall, I think the maps need some work. Please choose different colors for the LTC rate above and below symbology since you are using blue and red for the choropleth map.

-Lines 177-178: I don't think a a 0.86% increase between 2010-2018 is steady, nor notable. It's clearly stable. Suggest combining sentences 177-180 to show # of counties with high LTC decreased (significant difference?), but the overall state rate was stable.

-Please add "%" in the parentheses in Table 2 and a space between the n and (%).

-Could be worthwhile to produce a few bivariate maps of the LTC outcome and the covariates with the highest predictive power.

-Can you show a time series of new cases per year?

-Please discuss uncertainty of using county-level data, which does not accurately depict the spatial heterogeneity of care at smaller administrative units (think of the MAUP problem).

6. PLOS authors have the option to publish the peer review history of their article (what does this mean?). If published, this will include your full peer review and any attached files.

Reviewer #1: No

Reviewer #2: No

Reviewer #3: No

---

## [Author Response · Author response to Decision Letter 0]

16 Apr 2023

Reviewer #1

This is a well-written manuscript describing a county-level analysis of predictors of linkage to care among people living with HIV in South Carolina. Overall, the paper is clearly presented. I only have a few minor comments and questions.

1. The outcome of interest was a viral load or CD4 measurement within one month of HIV diagnosis. I recognize that this is based on federal recommendations, but at the very least a descriptive presentation of time to viral load or CD4 count would be useful. For example, how many occurred >30 days but <60 days or <90 days?

Response: Thank you for the suggestion. In the revision, we added a text description of the percentage of linkage to care within 30 days, 60 days and 90 days (Page 11; lines 175-179) with a bar chart illustrated in supplemental figure 2. 

S2 Fig. Linkage to care within 30 days, 60 days, or 90 days of HIV diagnosis in South Carolina, 2010 through 2018

2. Methods, line 99: should this be: "Successful linkage to care" divided by the number of newly diagnosed HIV cases? I.e., addition of 'newly' to this sentence?

Response: We agree and have added ‘newly’ to this sentence.

3. Results, lines 188-195: This paragraph is difficult to follow and would benefit from editing for clarity.

Response: We have rewritten this paragraph . Please refer to lines 181 to 191 in the manuscript. 

4. Table 3: Consider adding one to two decimal places to the count of Ryan White HIV centers and mental health centers per 100,000 people. Also, there is a minor typo here ("Pyan White" instead of "Ryan White").

Response: We added two decimal places as suggested, and we changed the typo of “Pyan White” to “Ryan White”. 

5. Results, lines 201-206: Lancaster and Sumter counties are repeated in this list.

Response: We deleted the repeated counties in the list.

6. The authors have used statistical methods to identify predictors of linkage to care; however, they have not conducted a causal analysis. Thus, use of words like 'determinants' is unwarranted. In most of the Discussion, 'predictors' is used. However, in the title and in the first paragraph of the Discussion, the word 'determinants' is used. The authors are encouraged to reconsider this word usage so as not to overstate the findings.

Response: We agreed and rephrased ‘determinants’ and ‘predictors’ into ‘Associations’ and ‘relationships’ throughout the manuscript.

Reviewer #2

Overall, the authors are bringing to light the social and environment factors that might be contributing to patterns in linkage to care in the state of South Carolina. This is helpful information to learn about, however, there are several items that have me hesitant to accept for publication. Please see below: 

1. Background

a. Lines 48 & 52: you state that PLWH and their linkage to care. Linkage to care is not based on persons living with HIV. It is based on persons newly diagnosed with HIV. Throughout the paper, you state PLWH and it should be newly diagnosed. PLWH are based on prevalence. That is, persons who are currently living with diagnosed HIV but could have been diagnosed at any time. The care outcomes for PLWH are receipt of care, retention in care, and viral suppression. https://www.cdc.gov/hiv/library/reports/hiv-surveillance/vol-27-no-3/index.html

Response: Thank you for pointing this out. We agreed and changed the description of people living with HIV (PLWH) into people newly diagnosed with HIV (PNWH) when talking about linkage to care throughout the manuscript. 

a. There are no mention of what HIV looks like in SC. How many cases on average annually? What is the distribution of cases by age, sex, race/ethnicity, transmission category? This information would be helpful to the reader to understand SC.

Response: We agreed and added the related description of HIV epidemiology in SC in the introduction part (page 2; lines 28 to 33). 

“According to the state surveillance data, there were around 748 PNWH annually in SC from 2009 to 2020. [5] Among them, men, African Americans, people aged 20-29, and men who have sex with men were disproportionately affected by HIV, making up low percentages of SC’s total population but comprising high percentages of PNWH. [5] For example, men comprise 48% of SC’s total population but makeup 80% of 1,556 PNWH in SC during the two-year period 2018-2019. [5]”

2. Methods

a. Line 83: eHARS is enhanced not electronic 

Response: We have changed ‘electronic’ into ‘enhanced’.

b. Lines 57 & 84: How do you calculate LTC rates? If LTC is a percentage? Also, you reference using CDC defn for LTC, you want to cite it here. If it’s the same as reference 4, then it is not clear

Response: We agreed that “LTC rate” is not the right term for our calculation, and we changed it to “LTC percentage” throughout the manuscript. According to CDC, timely LTC was measured by records of � 1 CD4 (count or percentage) or viral load tests performed within one month after HIV diagnosis, including tests performed on the same date as the date of diagnosis. [4] Based on this definition, we classified the individual-level LTC status as “timely LTC” and “delayed LTC.” The county-level timely LTC percentage was calculated as the number of “timely LTC” divided by the number of newly diagnosed HIV cases for each county in the specified calendar year. In addition, we cited the website of CDC’s definition of LTC as suggested. 

c. Line 86: you state them as “risk factors” however you are obtaining information such as gender, age, race, etc. I recommend not using the word “risk”. Some risk-related terms can be stigmatizing and may imply that the condition is inherent to a person or group rather than the actual causal factors. Could use “variables” or “factors” 

Response: We agreed and replaced ‘risk factors’ with ‘variables”. 

d. What 5-year estimates did you use from the American Community Survey? That is not clear.

Response: In ACS, it reported the 5-year estimates, which refer to data collected over the past five years for each calendar year from 2010 to 2018. For example, in 2018, the 5-year estimation refers to data collected from 2014 to 2018. We use different ACS dataset for different years. We clarified this issue in lines 110-112 and 193-198.

e. For all the data sources, ACS, County Health Rankings & Roadmaps, and US Congress joint Economic Committee, please include what year(s) of data you used.

Response: We added one column in table 1 to describe the year used for each variable in analyses to make it clear. 

f. Why were the variables “Male” and “Black” picked for predictors? 

Response: The percentage of Black residents, alternatively referred to as Black racial composition, has been suggested as a proxy measure of racial residential segregation among the Black population at the county level. A higher value of the Black racial composition aligns with more Black and non-Black residential segregation. The percentage of male persons may be related to masculinity norms in the local area, especially in the Deep South States. [42] Thus, we assumed counties with higher percentages of male or black residents were less likely to have high LTC percentages. That’s why we incorporate “Male” and “Black” into the model. 

g. Under the header County-level LTC status (lines 94-103): It appears that percentages, not rates, are calculated. So that readers are clear that your calculation aligns with CDC’s, I highly recommend using the same language as CDC when reporting on the outcome. Linkage to HIV medical care within 1 month after HIV diagnosis was measured by documentation of ≥ 1 CD4 (count or percentage) or viral load tests performed ≤ 1 month after HIV diagnosis, including tests performed on the same date as the date of diagnosis https://www.cdc.gov/hiv/library/reports/hiv-surveillance/vol-27-no-3/content/technical-notes.html. Also, linkage to care for national-level linkage to care data from CDC include different states for different years (ranging from 14 jurisdictions in 2010 to 43 jurisdictions in 2018 – see Technical Notes from NCHSSTP AtlasPlus here: https://gis.cdc.gov/grasp/nchhstpatlas/main.html). Therefore, it may not be accurate to call them national comparisons for the different years (and may need to state which jurisdictions are included in the “national level” for each year you are using). 

Response: As suggested, we modified and used the same language as CDC in the outcome definition part. Please refer to lines 77 to 79 in the un-tracked version manuscript. In addition, we agreed that it’s needed to state which jurisdictions were included. We described the jurisdictions in the methods part when mentioning the nation-level LTC (lines 86 to 91) and listed the detailed jurisdiction for each year in supplemental table 1. 

h. For Table 1: it would be helpful to include what year(s) for each variable. Also, table 1’s label states, “The detailed definition and cut-point of each county-level predictor”. However, I do not see any cutpoints in the table. 

Response: Thank you for the suggestion. We added one column in table 1 to describe the year used for each variable in analyses to make it clear. We did not use the cutoff in defining the variables and deleted it in the revision.

i. For the ACS data, how did the writer compute data for the 3 years: 2010, 2014, and 2018? Since the 5-year estimate is a period estimate, was the same ACS dataset used for the different years? If so, then that needs to be stated in the methods. 

Response: Different ACS data set is used for different year. In ACS, it reported the 5-year estimates refer to data collected over the past five years for each calendar year from 2010 to 2018. For example, in 2018, the 5-year estimation refers to data collected from 2014 to 2018. The 5-year estimated reported at the diagnosis year were used. Due to the limit of space, we only described the summary of 5-year estimates at year 2010, 2014, and 2018 in table 3. We clarified this issue in the methods part as suggested. Please refer to page 8, lines 110 -112 in the clean version.

j. Similar to the previous statement, it needs to be clear which years of each of the 25 variables was used.

Response: We added one column in table 1 to describe the year used for each variable.

k. Line 164: The statistical analysis section needs to provide more information on exactly how the analysis was computed for each for the 25 variables. Also, it is not clear how the data years (2010, 2014, and 2018) fit into the GEE model. More information is needed for clarity for the reader.

Response: We used longitudinal data from 2010 to 2018 to fit a Generalized Estimating Equation (GEE) model with the stepwise selection to explore the relationship between 25 county-level variables and LTC status. In the stepwise selection, we choose a p-value of 0.2 to select the variables. We modified the statistical analysis section to make the description clearer. Please refer to page 10, lines 162-168.

3. Results

a. Lines 177-180: “The state average LTC rate of SC steadily increased from 60.41% in 2010 to 61.27% in 178 2018, but still in a smaller magnitude than the increasing temporal trend at the national level. 179 And this explained why the number of counties with a high LTC decreased from 16 (34.8%) in 180 2010 to 8 (17.4%) in 2018 among 46 counties.” Again, this isn’t an apples-to-apples comparison. There are different numbers of jurisdictions in the national data for each year. 

Response: Agreed. We added the information of the jurisdictions in each year for the “national level” calculation in detail in the methods part. Please see below.

“According to the technical notes from CDC NCHHSTP AtlasPlus, national LTC is presented for persons aged � 13 years and only for states with complete laboratory data (at least 95% of laboratory results are reported to the surveillance programs and transmitted to the CDC). From 2010 to 2018, the calculation of national LTC percentage ranges from 14 to 43 jurisdictions. [24] The list of jurisdictions for which data are presented by year are presented in S1 Table.”

b. Throughout the results section, please determine if you will use one decimal point or 2. It is not consistent throughout.

Response: We decided to use two decimals. All results were updated throughout the paper for consistency. 

c. Table 3: again, it is not clear if you used the same dataset for the predictor variables for the different years. Also, Ryan White is misspelled in the table (it says Pyan White). 

Response: Except for adding one column in Table 1 describing the years for each variable, additional notes were made under Table 3 to make the variables and corresponding years of data used clear. In addition, we edited the typo of “Pyan White”.

d. Lines 200-206: There is discussion of counties in different regions (Upstate, Midlands, Pee Dee). This is the first time this information is mentioned. This type of information should be included in the Methods section. 

Response: We agreed and added the information in the description of Figure 1 in the statistical analysis. The information added is as below:

“The 46 counties in the nine maps were further grouped based on four Public Health Regions in SC, including Upstate, Midlands, Pee Dee, and Lowcountry. [36]”

e. Line 217: The CI should include ~ and not –

Response: Thank you for pointing this out. We changed ~ into – as suggested. 

f. Table 4: Some of the confidence intervals have ~ and others have , . Also in Table 4, please include a footnote/note at the bottom of the table stating that the results in bold are statistically significant. Also, what does “N/A” mean? It’s not clearly stated. 

Response: Thank you for pointing this out. We changed all “,” into “~ “ to make it look consistent throughout the table. We added footnote at the bottom of table to indicate that the results in bold are statistically significant and “-“ were variables not selected by stepwise selection in Table 4.

g. There is a concern with some counties having small numbers. The rule of thumb that CDC uses for HIV surveillance data is that results are based on stable numbers (i.e., based on 12 or more diagnoses). If they are less than 12, then they are not considered reliable. It appears there are some counties in your study that run into this issue

Response: We examined the new diagnosis over time using the GEE model which does not have any convergence issue in our study. Thus, we have less concern about the modelling approach. We acknowledge that there are some counties that run into the issue of having less 12 newly diagnosed cases in some years, and we pay caution in the interpretation. 

h. Also, there may be some counties with small population sizes. What that taken into account? 

Response: We agreed that the population size might influence the results and incorporated population size as one variable in the regression model.

i. For the results, did the writers take into account the number of HIV diagnosis by specific demographics? You results show that males and Black persons were statistically significant findings, but your model didn’t take into account the number of HIV diagnoses that are Black and that are males. From NCHHSTP AtlasPlus, it shows that in just in 2018 alone, here are the numbers and rates for the top 3 racial/ethnic groups:

Race Cases Rate per 100,000 Population

Black/African American 448 40.0 1,119,585

White 172 6.1 2,815,183

Hispanic/Latino 56 25.9 216,019

 Additionally, there are more males being diagnosed in SC:

Sex Cases Rate per 100,000 Population

Male 564 27.3 2,063,390

Female 161 7.2 2,232,959

Wouldn’t this have some impact on the results? Wouldn’t this introduce some type of bias into your analysis? Also, of those cases diagnosed, what percentage were linked to care for the demographic information? This information would be needed to tell the full picture. 

Response: We agreed that this might impact the results and we added the percentage of male persons and Black persons among people newly diagnosed with HIV in each year from 2010 to 2018 into the analysis. We described the percentage of male and Black persons among people newly diagnosed in Table 3 and added them as county-level variables in the GEE model.

4. Discussion

a. Lines 243-246: For instance, Allendale is a county located in the Lowcountry area with a consistently low LTC rate. This may be because it has the largest proportion of the Black population, the highest percentage of unemployment/poverty, and the lowest median income.” Language that implicitly contains a negative judgement about the character of a person or a group of people (especially the statement about “the Black population”). It also may blame people for circumstances beyond their control. Such language often contributes to disapproving views of, or discrimination against, a group of people. I recommend making sure the statements are written as to not place blame, stigmatize, or offend readers. 

Response: We agreed and re-wrote the whole paragraph to avoid placing blame. Please refer to lines 238 to 247 in the clean version manuscript.

b. For the discussion on counties, there needs to be more understanding of these regions and counties. Readers outside of SC (and even those inside SC), may not know the distribution of population characteristics for the state. Therefore, further explanation is needed. The geospatial work could actually be a paper in and of itself.

Response: Thank you for pointing this out. We agreed and added some brief discussions on those regions and counties. Please refer to lines 241 to 247 in the clean version manuscript.

c. Lines 250-256: You found a negative association between % Black and high LTC. Again, couldn’t some of this be explained because most newly diagnosed cases are among Black persons? The study you mention from Florida is a good suggestion for your analysis. Using a multi-level (or HLM) modeling to that take into account both individual and group level factors. 

Response: We acknowledge that it’s very valuable to conduct the multilevel analysis. However, the scope of this paper is to investigate the percentage of linkage to care at the county level and identify the county that could be the target of intervention in the future. In the multilevel analysis, the individual linkage to care status should be used as the outcome, which cannot address our focus on the percentage of linkage to care at the county level. In the revision, we incorporate more information aggregated from the individual level (e.g., percentage of Black persons among people newly diagnosed with HIV) into our analysis to address the issue that most newly diagnosed cases are among Black persons.

d. Lines 257-265: Similar to above, your study found a negative association between males and high LTC. Again, couldn’t some of this be explained by most cases being among males? 

Response: Same as above, we added the percentage of male persons among people newly diagnosed with HIV in each year from 2010 to 2018 into analysis to address the issue that most newly diagnosed cases are among male persons.

e. Lines 286 and 294: the fact that the numbers are so small, these results may be very unstable. There should be caution when presenting results from such case count. Therefore, there should be no discussion on these results. 

Response: Thank you for this suggestion. We agreed and deleted related discussions if there were a small number issue. 

5. Overall Recommendations

a. Background needs to be reworked based on recommendations above

Response: Thank you for the recommendation. We have reworked the background as suggested.

b. Methods needs to include pertinent information that is recommended above

Response: Thank you for the recommendation. We have included the information suggested in the methods part. 

c. The analysis needs to be reconsidered. Suggestion is to run a multilevel analysis that looks at individual and group level data 

Response: Thank you for the recommendation. We acknowledge that it’s very valuable to conduct the multilevel analysis. However, the scope of this paper is to investigate the rate of linkage to care at the county level and identify the county that could be the target of intervention in the future. In the multilevel analysis, the individual linkage to care status should be used as the outcome, which cannot address our focus on the percentage of linkage to care at the county level. In the revision, we incorporate more information generated from the individual level (e.g., percent of Black persons among people newly diagnosed with HIV) into our analysis. 

d. Proceed with caution when discussing numbers less than 12

Response: Thank you for the recommendation. We agreed and deleted discussion regarding numbers less than 12. 

e. Overall, several concerns with the analysis and results (and interpretation of some of the results)

Response: Thank you for the recommendation. We modified the analysis, results, and interpretation as suggested. 

f. Be aware of language that is stigmatizing

Response: Thank you for the recommendation. We modified the language that is stigmatizing.

Reviewer #3

This manuscript examines factors that influence linkage to HIV care in South Carolina at the county level between 2010 and 2018. The paper is very well written and clearly and concisely summarized. Overall, the study is well done and don't have any major concerns, but numerous minor issues should be addressed before publication consideration.

Minor comments:

1. Please provide the IRB IDs approved bu SC and SC DHEC in text (line 93).

Response: This study was approved by the USC and SC DHEC IRB (#Pro00068124) in 2017. We added the IRB IDs as suggested. 

2. Please improve the resolution of figures 1 and 2.

Response: Thank you for pointing this out. We have improved the resolution of figures 1 and 2 as suggested.

3. Figure 1: Please add a white halo around the labels in the 2010 map. Overall, I think the maps need some work. Please choose different colors for the LTC rate above and below symbology since you are using blue and red for the choropleth map.

Response: We added the white halo around the labels, and chose different colors for the LTC percentage above and below symbology as suggested. 

4. Lines 177-178: I don't think a a 0.86% increase between 2010-2018 is steady, nor notable. It's clearly stable. Suggest combining sentences 177-180 to show # of counties with high LTC decreased (significant difference?), but the overall state rate was stable.

Response: We edited this part to “The number of counties with a high LTC decreased from 34 (73.91%) in 2010 to 21 (45.65%) in 2018 among 46 counties. However, the state average timely LTC percentage in SC is relatively stable, with the LTC percentage being 78.55% in 2010 and 80.99% in 2018.”

5. Please add "%" in the parentheses in Table 2 and a space between the n and (%).

Response: We added % and the space as suggested.

6. Could be worthwhile to produce a few bivariate maps of the LTC outcome and the covariates with the highest predictive power.

Response: We agreed and added a bivariate choropleth map illustrating the spatial distribution of the LTC outcome and the covariates with the highest predictive power, which is the number of mental health centers per people newly diagnosed with HIV in the current study. Please refer to S3 Fig. 

7. Can you show a time series of new cases per year?

Response: We added a bar and line chart to illustrate the time series of new cases per year as suggested. Please check S1 Fig. 

Supplemental Figure 1 The number of newly diagnosed HIV cases in South Carolina, 2010 through 2018

8. Please discuss uncertainty of using county-level data, which does not accurately depict the spatial heterogeneity of care at smaller administrative units (think of the MAUP problem).

Response: Thanks for the comment. We discuss the uncertainty of using county-level data in the limitation part as suggested. Please check below:

Fourth, there may be a modifiable areal unit problem (MAUP) since county-level data were used in the analysis. We need to be cautious when generating findings of the current study to other administrative units, such as census tract.

Editor’s comments and additional requirements:

Response: Thank you for the comment. We formatted the style of this manuscript and renamed the files based on PLOS one’s submission guideline as required.

Response: The IRB approved this study as a non-human subject study, and no participant consent is needed. Additional information was added to the methods part as suggested. 

“The research reported in this publication was supported by the National Institute of Allergy and Infectious Diseases of the National Institutes of Health under Award Number R01AI127203 and R01AI164947. This work was also partially supported by a SPARC Graduate Research Grant from the office of the Vice President for Research at the University of South Carolina (grant #: 115400-22-59203). Dr. Xueying Yang’s effort is supported by ASPIRE -I, TRACK-2 from the office of the Vice President for Research at the University of South Carolina (grant #: 115400-22-60028). The content is solely the responsibility of the authors and does not necessarily represent the official views of the National Institutes of Health. Both NIAID and NIH had no role in the design of the study, collection, analysis, and interpretation of the data.”

“The research reported in this publication was supported by the National Institute of Allergy and Infectious Diseases of the National Institutes of Health under Award Number R01AI127203 (PI: XL) and R01AI164947 (PI: JZ,BO). This work was also partially supported by a SPARC Graduate Research Grant from the office of the Vice President for Research at the University of South Carolina (grant #: 115400-22-59203) (PI: FS). Dr. Xueying Yang’s effort is supported by ASPIRE -I, TRACK-2 from the office of the Vice President for Research at the University of South Carolina (grant #: 115400-22-60028). The content is solely the responsibility of the authors and does not necessarily represent the official views of the National Institutes of Health. The funders had no role in study design, data collection and analysis, decision to publish, or preparation of the manuscript.”

Response: Thank you for pointing this point. We revised the acknowledgement part into “The authors thank the SC Department of Health and Environmental Control (DHEC), the office of Revenue and Fiscal Affairs (RFA), and other SC agencies for contributing the data in South Carolina”. 

The amended funding statements are “The research reported in this publication was supported by the National Institute of Allergy and Infectious Diseases of the National Institutes of Health under Award Number R01AI127203 (PI: XL) and R01AI164947 (PI: JZ,BO). This work was also partially supported by a SPARC Graduate Research Grant from the office of the Vice President for Research at the University of South Carolina (grant #: 115400-22-59203) (PI: FS). Dr. Xueying Yang’s effort is supported by ASPIRE -I, TRACK-2 from the office of the Vice President for Research at the University of South Carolina (grant #: 115400-22-60028). The content is solely the responsibility of the authors and does not necessarily represent the official views of the National Institutes of Health. The funders had no role in study design, data collection and analysis, decision to publish, or preparation of the manuscript.” We added the amended statement in the cover letter. 

Response: Data is not publicly available due to provisions in our data use agreements with state agencies/data providers, institutional policy, and ethical requirements. We make access to such data available via approved data access requests from the IRB of the University of South Carolina (contact Lisa M. Johnson at lisaj@mailbox.sc.edu).

We added the data availability statement in the cover letter as required. Please check it out. Thanks!

5. We note that you have indicated that data from this study are available upon request. PLOS only allows data to be available upon request if there are legal or ethical restrictions on sharing data publicly. For more information on unacceptable data access restrictions, please see http://journals.plos.org/plosone/s/data-availability#loc-unacceptable-data-access-restrictions

Response: We added the data availability statement in the cover letter as required. Please check it out. Thanks!

Response: The Institutional Review Boards at the University of South Carolina and the SC Department of Health and Environmental Control approved the study protocol; the IRB number is #Pro00068124. The IRB approved this study as a non-human subject study, and no participant consent is needed. We added this information io the methods part. Please refer to lines 72 to 75 in the clean version manuscript. Thanks!

7. We note that [Figure 1) #] in your submission contain [map/satellite] images which may be copyrighted. All PLOS content is published under the Creative Commons Attribution License (CC BY 4.0), which means that the manuscript, images, and Supporting Information files will be freely available online, and any third party is permitted to access, download, copy, distribute, and use these materials in any way, even commercially, with proper attribution. For these reasons, we cannot publish previously copyrighted maps or satellite images created using proprietary data, such as Google software (Google Maps, Street View, and Earth). For more information, see our copyright guidelines: http://journals.plos.org/plosone/s/licenses-and-copyright.

Response: Thank you for the comments. We have double-checked the copyright issue, and we are confident that no copyright materials were used in the maps.

---

## [Decision Letter · Decision Letter 1]

18 May 2023

County-level variations in linkage to care among people newly diagnosed with HIV in South Carolina: A longitudinal analysis from 2010 to 2018

PONE-D-22-15791R1

Dear Dr. Fanghui Shi,

We’re pleased to inform you that your manuscript has been judged scientifically suitable for publication and will be formally accepted for publication once it meets all outstanding technical requirements.

Kind regards,

Csaba Varga, DVM MSc PhD

Academic Editor

PLOS ONE

Reviewer #1: All comments have been addressed

Reviewer #3: All comments have been addressed

2. Is the manuscript technically sound, and do the data support the conclusions?

Reviewer #1: Yes

Reviewer #3: Yes

3. Has the statistical analysis been performed appropriately and rigorously? 

Reviewer #1: Yes

Reviewer #3: Yes

4. Have the authors made all data underlying the findings in their manuscript fully available?

Reviewer #1: (No Response)

Reviewer #3: No

5. Is the manuscript presented in an intelligible fashion and written in standard English?

Reviewer #1: Yes

Reviewer #3: Yes

6. Review Comments to the Author

Reviewer #1: (No Response)

Reviewer #3: The authors have addressed my comments and the manuscript appears to be suitable for publication. However, the bivariate map legend should be improved to include distinct class break values for each axis.

7. PLOS authors have the option to publish the peer review history of their article (what does this mean?). If published, this will include your full peer review and any attached files.

Reviewer #1: No

Reviewer #3: No

---

## [Editor Report · Acceptance letter]

22 May 2023

PONE-D-22-15791R1 

County-level variations in linkage to care among people newly diagnosed with HIV in South Carolina: A longitudinal analysis from 2010 to 2018 

Dear Dr. Shi:

I'm pleased to inform you that your manuscript has been deemed suitable for publication in PLOS ONE. Congratulations! Your manuscript is now with our production department. 

Kind regards, 

on behalf of

Dr. Csaba Varga 

Academic Editor

PLOS ONE